# At the Crossroads of Apoptosis and Autophagy: Multiple Roles of the Co-Chaperone BAG3 in Stress and Therapy Resistance of Cancer

**DOI:** 10.3390/cells9030574

**Published:** 2020-02-28

**Authors:** Donat Kögel, Benedikt Linder, Andreas Brunschweiger, Silvia Chines, Christian Behl

**Affiliations:** 1Experimental Neurosurgery, Neuroscience Center, Goethe University University Hospital, D-60590 Frankfurt am Main, Germany; linder@med.uni-frankfurt.de; 2German Cancer Consortium (DKTK), Partner Site Frankfurt, D-60590 Frankfurt am Main, Germany; 3Faculty for Chemistry and Chemical Biology, TU Dortmund University, 44227 Dortmund, Germany; andreas.brunschweiger@tu-dortmund.de (A.B.); silvia.chines@tu-dortmund.de (S.C.); 4Institute of Pathobiochemistry, University Medical Center, Johannes Gutenberg-University, D-55099 Mainz, Germany; cbehl@uni-mainz.de

**Keywords:** cancer, therapy resistance, chaperones, apoptosis, cell survival, autophagy, tumor progression

## Abstract

BAG3, a multifunctional HSP70 co-chaperone and anti-apoptotic protein that interacts with the ATPase domain of HSP70 through its C-terminal BAG domain plays a key physiological role in cellular proteostasis. The HSP70/BAG3 complex determines the levels of a large number of selective client proteins by regulating their turnover via the two major protein degradation pathways, i.e. proteasomal degradation and macroautophagy. On the one hand, BAG3 competes with BAG1 for binding to HSP70, thereby preventing the proteasomal degradation of its client proteins. By functionally interacting with HSP70 and LC3, BAG3 also delivers polyubiquitinated proteins to the autophagy pathway. BAG3 exerts a number of key physiological functions, including an involvement in cellular stress responses, proteostasis, cell death regulation, development, and cytoskeletal dynamics. Conversely, aberrant BAG3 function/expression has pathophysiological relevance correlated to cardiomyopathies, neurodegeneration, and cancer. Evidence obtained in recent years underscores the fact that BAG3 drives several key hallmarks of cancer, including cell adhesion, metastasis, angiogenesis, enhanced autophagic activity, and apoptosis inhibition. This review provides a state-of-the-art overview on the role of BAG3 in stress and therapy resistance of cancer, with a particular focus on BAG3-dependent modulation of apoptotic signaling and autophagic/lysosomal activity.

## 1. Introduction

The multifunctional HSP70 co-chaperone and anti-apoptotic protein BAG3 (also called Bis [BCL-2 interacting death suppressor] and CAIR-1) is a well investigated member of the Bcl-2-associated athanogene (BAG) protein family [1,2,3]. This highly conserved family of co-chaperones [4] interacts with the ATPase domain of heat shock protein 70 (HSP70) through the so-called BAG domain, a C-terminal structural domain consisting of three anti-parallel α-helices (Figure 1). Since a crystal structure of the BAG3-HSP70 complex has to date not been published, we used the crystal structure of the homologous BAG1-HSP70 complex (PDB: 4HWI, Figure 1a; [5]) to create a homology model. The primary sequence of BAG3 was retrieved from the Uniprot database [6] and a homology model was computed with SwissModel (Figure 1b) [7]. The superimposition is visualized in Figure 1c by Pymol (The PyMOL Molecular Graphics System, Version 2.0 Schrödinger, LLC): the two proteins overlap except for a few amino acid residues distal to the protein-protein interaction. Plausible key interactions between the two proteins were highlighted and analyzed by visual inspection. Salient interactions in the homology model are salt bridges, such as those between Arg480 (BAG3) and Asp285 (HSP70), and between Asp456 (BAG3) and Arg262 (HSP70) (Figure 1d). Similar salt bridges have also been observed in the X-ray structure of the BAG1-Hsc70 complex [4].

In addition to the BAG domain, BAG3 also contains a WW domain, a proline-rich (PxxP) domain and two conserved IPV motifs, one located at the center and one near the N-terminus of the protein. The complex between BAG3 and HSP70 selectively binds to a wide variety of client proteins. Since BAG3 competes with BAG1 for binding to the ATPase domain of HSP70, BAG3 can interfere with the HSP70/BAG1-dependent delivery of clients to proteasomal degradation. In concert with HSP70 and LC3, BAG3 also can target polyubiquitinated client proteins to autophagic degradation. Therefore, BAG3 is centrally involved in regulation of both major cellular protein degradation pathways, proteasomal degradation, and autophagy, and plays a key physiological role in cellular proteostasis [1,2,3]. By regulating the protein levels of distinct client proteins in this manner, BAG3 is able to modulate a wide variety of physiological processes, including apoptosis, development, and cytoskeletal dynamics/organization. In addition to its physiological function, BAG3 is also involved in several pathological conditions, including cardiomyopathies, age-related neurodegenerative diseases, and cancer. In malignant diseases, BAG3 mostly exerts oncogenic functions and is known to regulate several key hallmarks of cancer, including cell survival, cell adhesion, metastasis, and angiogenesis [2].

The human BAG3 gene is located on the long arm of chromosome 10 (10q26.11). Most non-malignant cell types express only low basal levels of BAG3, while BAG3 is constitutively expressed in cardiac myocytes and skeletal muscle cells. However, BAG3 expression is induced by various stress stimuli such as oxidative stress, heat, heavy metals, virus infection, proteasome inhibition, and serum deprivation [3]. The transcription factor heat shock factor 1 (HSF1), which acts upstream from BAG3 and HSP70, is the major driver of BAG3 induction under stress conditions. Basal BAG3 expression was found to be elevated in many cancer entities, e.g., in lymphocytic leukemia, neuroblastoma, glioma, thyroid, breast, and pancreatic cancer [2] (Table 1). Since BAG3 is capable of promoting cell survival signaling by interacting with distinct client proteins in complex with HSP70, this BAG3 overexpression in general contributes to apoptosis resistance of the tumors [8].

Apoptosis is characterized by a cascade of molecular events that are initiated by distinct upstream signals and culminate in the activation of effector caspases, the major executors of apoptotic cell death [9]. Resistance to cell death caused by defects in apoptotic pathways and overexpression of anti-apoptotic proteins is a general hallmark of cancer. Pro- and anti-apoptotic members of the Bcl-2 family are well-established key regulators of apoptotic cell death. The Bcl-2 family proteins can be classified into three subfamilies: (i) the pro-apoptotic BH3-only proteins, which have only one domain (BH3 domain) in common; (ii) the pro-apoptotic Bax-like proteins, which contain three such domains (BH1,2,3); and (iii) the anti-apoptotic Bcl-2-like proteins, which contain four homology domains (BH1-4). Bcl-2 family proteins play a pivotal role in regulation of the intrinsic (mitochondrial) pathway of apoptosis that triggers Bax/Bak-dependent mitochondrial outer membrane permeabilization (MOMP) and the release of pro-apoptotic factors into the cytoplasm. This intrinsic apoptosis pathway is kept in check by the pro-survival Bcl-2 family members (Bcl-2, Bcl-xL, Mcl-1, Bcl-w and Bfl-1) that are often overexpressed in cancer, a phenomenon that is driven by several distinct mechanisms, including chromosomal translocation, enhanced mRNA expression, and protein stability. A key feature promoting the anti-apoptotic function of BAG3 is represented by its stabilizing effect on the pro-survival Bcl-2 family proteins, including Bcl-2, Bcl-xL, and Mcl-1, a BAG3-driven event supporting the apoptosis-antagonizing function of these proteins. The HSP70/BAG3 complex with anti-apoptotic Bcl-2 proteins is formed to restrict proteasomal turnover of these proteins, leading to an enhanced anti-apoptotic capacity and prevention of apoptosis induction. Mechanistically, BAG3 was shown to interfere with the HSP70-mediated delivery of the anti-apoptotic proteins Mcl-1, Bcl-2, and Bcl-xL to the proteasomal pathway by competing with BAG1, a protein that also functions as a co-chaperone of HSP70. In the case of Mcl-1, the BAG3 interaction with Mcl-1 and HSP70 may prevent or destabilize the binding of Mcl-1 to HSP70, reducing the delivery of Mcl-1 to the proteasome and supporting the anti-apoptotic activity of Mcl-1.

Pro-survival Bcl-2 family members are highly relevant targets for cancer therapy and several Bcl-2 antagonists, also termed BH3 mimetics, have been developed at the preclinical and clinical stages. The apoptopsis-inducing BH3 mimetics ABT-263 (Navitoclax) and its analog ABT-737 are known to target Bcl-2 and Bcl-xL with high affinity, but not the structurally distinct family member Mcl-1. Boiani et al. used neuroblastoma cells as a model to demonstrate that BAG3 protects Mcl-1 from proteasomal degradation, thereby promoting its anti-apoptotic activity. Depletion of BAG3 led to a marked reduction in Mcl-1 protein levels and was able to overcome ABT-737 resistance. This study identified BAG3 as a potential target for combined cancer therapies with Bcl2-antagonists [10].

In addition to its effects on the pro-survival Bcl-2 proteins, BAG3 can also activate the anti-apoptotic NF-*κ*B pathway, in this case by protecting IKK-*γ* (NEMO), a subunit of the IκB kinase complex, from proteasomal delivery, thereby causing sustained activation of NF-*κ*B and cell survival [11]. Furthermore, the levels of survivin, a pro-survival protein belonging to the IAP (inhibitor of apoptosis) family, were reported to be downregulated after BAG3 and HSP70 depletion, pointing to a stabilization of survivin by the HSP70/BAG3 complex [12]. Indeed, the work by Colvin et al. proposed that HSP70-BAG3 interactions regulate several additional cancer-related signaling networks and may represent a hub molecule in cancer cell signaling. Accordingly, it was shown that interaction of (the PxxP region of) BAG3 with the SH3 domain of Src mediates the effects of HSP70 on Src signaling. The authors also found that the HSP70-BAG3 module modulates the activity of several key drivers of cancer, including the transcription factors FoxM1, Hif1α (in addition to NF-κB), the translation regulator HuR, and the cell-cycle regulator p21, with possible implications for tumor cell survival, cellular metabolism, stemness, and proliferation.

BAG3 is also involved in regulation of selective (macro)autophagy for the degradation of damaged client proteins known to accumulate under conditions of cellular stress. Macroautophagy, characterized by formation of autophagosomes and their subsequent fusion with lysosomes, is an evolutionarily conserved process in which cellular constituents are delivered to the autophagosomal-lysosomal pathway for bulk degradation [13]. Other general forms of autophagy are microautophagy and chaperone-mediated autophagy (CMA). We have previously demonstrated that BAG3 expression is inversely related to BAG1 expression under acute oxidative or proteasomal stress and during cell aging (“BAG1-BAG3-Expression Shift”). Under physiological conditions, the co-chaperone BAG1 interacts with HSP70 to transfer polyubiquitinated proteins to the proteasome. When misfolded and aggregated proteins accumulate under stress conditions, a switch from BAG1 to BAG3 expression ensures sustained intracellular proteostasis by recruiting the selective macroautophagy pathway via formation of a multichaperone protein complex containing BAG3, HSPB8, and HSP70 controlling selective degradation of protein substrates and stabilizing protein homeostasis [14,15,16,17,18,19,20]. BAG3-dependent degradation of misfolded client proteins via the autophagosomal/lysosomal pathway involves their retrograde transport along microtubules and the formation of a perinuclear cell compartment, the so-called aggresome in which misfolded client proteins are accumulated before being targeted for lysosomal degradation [18,21]. This BAG3-mediated aggresome-targeting requires the interaction of the PxxP region of BAG3 with the motor protein dynein [18,22]. Furthermore, the 14-3-3γ protein binds to the phosphoserine-containing 14-3-3 binding motifs RSQS136 and RSQS173 of BAG3 and to the dynein-intermediate chain (DIC), thereby acting as an additional molecular adaptor in BAG3-dependent aggresome-targeting [23].

The homeostatic, cytoprotective function of BAG3-dependent autophagy so far has mostly been studied in models of neurodegeneration and its possible role in cancer is only beginning to emerge [24,25,26]. Since cellular stress induced by cancer drugs and radiation leads to accumulation of damaged proteins, it is however likely that BAG3-dependent recruitment of the macroautophagy pathway and consolidation of cells via decreased proteotoxicity may additionally protect many tumors against apoptosis on top of the selective modulation of apoptosis regulators by BAG3 and the proteasomal pathway, as outlined above. In general, an extensive cellular crosstalk between the apoptosis and autophagy pathways involving dual regulators of both pathways (e.g., ATG5, ATG12, p53) is well established [27,28]. Therefore, the key functions of BAG3 in regulation of apoptosis and autophagy suggest that BAG3 may also play a role in the crosstalk between these two pathways, although the exact mechanisms linking apoptosis and autophagy signaling at the level of BAG3 remain to be defined. Since BAG3 is a mutimodal hub molecule (Figure 2), it could also be argued that its roles in modulation of apoptosis and autophagy are separate, distinct functions of the protein promoted by its specific interactors (Figure 2). One could hypothesize that the anti-apoptotic vs. pro-autophagic activity of BAG3 is driven by the respective cellular demand. Under conditions of severe misfolded protein accumulation (e.g., induced by proteasome inhibitors such as Bortezomib), there may be a shift from BAG3-dependent apoptosis inhibition towards more autophagy activation, possibly involving most/all of the cellular pool of BAG3 in this setting. Therefore, different types of cancer therapy may evoke distinct impacts on the BAG3-driven stress response regarding apoptosis vs. autophagy modulation. Future investigations on the molecular events involved in BAG3-dependent regulation of the crosstalk between apoptosis and autophagy will be crucial to better understand the responses of individual cancers to therapy.

On top of its roles in apoptosis and autophagy, BAG3 is also involved in modulation of other cellular functions with relevance for malignant diseases, including cell cycle progression, cellular metabolism, and cell adhesion. Accordingly, BAG3 was proposed to play a pivotal role during mitosis, where it functions as part of the protein quality control during mechanical strain and colocalizes with HSPB8 and p62 as a hyperphosphorylated form during mitosis. Fuchs et al. [29] showed that BAG3 is required for correct spindle assembly and orientation during mitosis, including the assembly of retraction fibers, which are thought to provide the necessary forces for spindle orientation. Interestingly, siRNA-mediated depletion of p62 could mimic these findings, thereby providing more evidence for a novel chaperone complex that is forming during mitosis.

Recently, it was also shown that binding of BAG3 to the enzyme glutaminase (GLS), which deaminates glutamine to initiate glutaminolysis, enhances autophagy [26]. Ectopic expression of BAG3 resulted in a strong increase in autophagy independently of Beclin or PI3K; instead, glutaminolysis, the breakdown of glutamine to feed cancer cell growth [30], was increased. The binding of BAG3 to GLS prevented its proteosomal degradation, whereas autophagy was hypothesized to be induced via ammonia that is derived from glutaminolysis, as was shown previously [31], indicating that BAG3 is a central factor bridging cancer metabolism and autophagy.

## 2. BAG3 in Stress and Apoptosis Resistance of Different Tumor Entities

### 2.1. Breast Cancer

BAG3 is frequently overexpressed in breast cancer and high BAG3 expression levels are correlated with a poor prognosis [32]. Furthermore, our own work demonstrated that BAG3 expression levels are correlated with chemotherapy resistance of triple-negative breast cancer (TNBC) [33], a primary obstacle for the treatment of these tumors. We applied derivatives of BT-549 and MDA-MB-468 TNBC cells adapted to growth in the presence of either 5-Fluorouracil, Doxorubicin or Docetaxel that exhibited enhanced cross resistance to chemotherapeutic drugs and decreased apoptosis sensitivity. In line with the observations of Boiani et al., lentiviral depletion of BAG3 evoked a robust downregulation of the pro-survival Bcl-2 family proteins Mcl-1, Bcl-2, and Bcl-xL, and restoration of drug-induced apoptosis, effects that were mimicked with the HSP70/BAG3 interaction inhibitor YM-1 and by KRIBB11, a selective transcriptional inhibitor of HSF-1. These data suggest that the increased apoptosis resistance of chemotherapy-adapted vs. completely untreated chemonaïve cells is functionally linked to BAG3-dependent stabilization of Bcl-2, Bcl-xL and Mcl-1. Furthermore, BAG3 depletion caused reduced cell adhesion and reverted the epithelial to mesenchymal transition (EMT)-like transcriptional changes observed in the BAG3-proficient, chemotherapy resistant cells. Based on these findings, BAG3 represents an interesting target to overcome therapy resistance of TNBC.

In line with these findings, Shields et al. reported that BAG3 is highly expressed in a subset of TNBC cell lines and primary tumor samples, while high expression of BAG3 in TNBC patients was associated with reduced recurrence-free survival, supporting the impact of BAG3 as an adverse prognostic factor [34]. The authors also observed a positive correlation between BAG3 and EGFR expression in TNBC cells and a direct interaction between BAG3 and EGFR signaling network components using a mass spectrometry approach. In TNBC cells, the combined targeting of BAG3 and EGFR was superior to inhibition of EGFR with Cetuximab alone, again highlighting the potential of BAG3 as a therapeutic target in TNBC [34].

Consistent with the findings of Das et al., an earlier study by Pasilass et al. applied a quantitative proteomic approach in the context of therapy-induced senescence to discover that BAG3 is upregulated after adriamycin treatment in MCF7 breast cancer cells. The authors also found a novel interaction between BAG3 and Major Vault Protein (MVP) that contributes to apoptosis resistance by activating the extracellular signal-regulated kinase1/2 (ERK1/2) pathway and proposed a model in which BAG3 binds to MVP and facilitates MVP accumulation in the nucleus, which sustains ERK1/2 activation [35].

Cancer stem cells represent a dynamic subpopulation of highly tumorigenic cells with stem cell properties that are held responsible for therapy resistance and tumor relapses. Liu et al. reported that BAG3 is induced under culture conditions that enrich breast cancer stem cell (BCSC)-like cells [32]. Ectopic BAG3 overexpression increased the percentage of CD44^+^/CD24^−^ CSC cells in the cultures, upregulated CXCR4 expression, and enhanced mammosphere formation, suggesting that BAG3 promotes CSC self-renewal and maintenance. The authors demonstrated a direct BAG3 interaction with CXCR4 mRNA, promoting CXCR4 expression via its coding and 3’-untranslational regions and proposed that BAG3 drives the BCSC-like phenotype through CXCR4 via interaction with its transcript. Collectively, the data support the notion that BAG3 is a potential therapeutic target of breast cancer.

Approximately 70% of newly diagnosed breast cancers are positive for expression and function of estrogen receptors (ER) and here the ER subtype ERalpha is predominantly expressed in human breast tumors. Consequently, in breast tumor tissue positive for ERalpha, this particular hormone receptor is one key pharmacological target. Since a pivotal role for downstream ER signaling in breast cancer is widely accepted, an anti-hormone therapy (e.g., with anti-estrogens or aromatase inhibitors) to substantially block ER function is frequently employed in the clinical setting. Breast cancer cells often display therapy resistance and it was suggested that an upregulated autophagic-lysosomal activity (autophagy) might facilitate and mediate this resistance and autophagy may represent a therapeutic target [36]. To get a clearer picture on the changes in key autophagy modulators and the general autophagic activity in breast cancer cells expressing ERs, autophagy-related expression profiles were established in MCF-7 breast cancer cells in vitro and verified in human breast tumor tissue samples. Interestingly it was found that ERalpha presence enhances autophagic activity in a non-canonical manner in vitro and in human tissue; obviously, here as well, BAG3 was functionally involved by mediating BAG3-dependent selective macroautophagy [24]. Taken together with the findings by Das et al. (see above) showing that BAG3 overexpression is correlated with cytoprotective autophagy, ultimately mediating resistance to cell death in selected chemoresistant breast cancer cells [33], there is good collective evidence that BAG3 plays a role in apoptosis resistance of breast cancer cells, very likely via both its anti-apoptotic and pro-autophagic activity. Consequently, a co-administration of BAG3 expression and function modulators may be applicable as a novel therapy path in the future.

In line with these observations made in breast cancer models, it was described previously for neuronal cells that Caspase 3, BAG1, and BAG3 are key targets of ERα [37] and in a model of oxidative stress resistance of neuronal cells. Again, BAG3 was found to take part in the resistance phenotype. Mouse clonal hippocampal cells (cell line HT22) that were selected for their stable resistance to oxidative stress, as executed by H_2_O_2_, showed massive changes in lysosomal and autophagic activity during redox adaptation [38]. Excitingly, again, a significant upregulation of BAG3 expression and a subsequently enhanced activity of the BAG3-mediated selective autophagy pathway was found. In addition to BAG3, changes in mitochondrial dynamics and protein expression, as well as other cell death-associated protein levels were significantly altered.

### 2.2. Leukemia

BAG3 was also shown to be increased in samples from chronic lymphocytic leukemia (CLL) patients in comparison to healthy B cells. Additionally, a shorter overall survival was observed in patients with high BAG3 expression. Knockdown of BAG3 in primary samples from the same study resulted in decreased levels of Bcl-2 in line with an increased apoptosis and migration rate [39].

Using human leukemia cell lines and nude mice xenografts, Liu et al showed that BAG3 expression was increased after inhibition of the proteasome using Bortezomib. In order to prevent this increase, BAG3 was silenced using a lentiviral shRNA-knockdown, which resulted in apoptosis-induction after bortezomib treatment in vitro and prolonged survival in vivo [40]. Furthermore, diethylmaleate-induced ROS-formation combined with BAG3-knockdown (siRNA) induced apoptosis in vitro in leukemia cells, but also in PBMCs [41]. These studies nicely illustrate that BAG3 is an important protein for cancer cell survival, but due to its role in stress responses, BAG3-inhibition likely will also have a profound impact on non-cancerous cells (e.g., PBMCs). These considerations should be taken into account when developing therapeutics (see below) against BAG3.

### 2.3. Colon Cancer

Using a cohort of Chinese patients, Shi et al. determined that BAG3 mRNA is higher expressed in tumor samples compared to normal tissue. Additionally, BAG3 expression correlates with tumor stage (TNM grading), differentiation, and metastasis. Using cell lines, the authors further determined that transient depletion of BAG3 using siRNA induced cell cycle arrest, which was accompanied by decreased expression of cyclins D1, A2 and B1. Furthermore, a reversal of EMT was observed in line with reduced invasion in vitro [42]. Another study analyzing a Chinese cohort of colorectal cancer patients also revealed that BAG3 is higher expressed in tumor compared to non-tumor adjacent tissue. Furthermore, tumors larger than five cm in diameter displayed higher BAG3 expression and, for the first time, BAG3 expression was found to be higher in tumors from females compared to male tumors, which might be related to the gender-dependent expression and function of the ER. The same study analyzed the effect of BAG3 overexpression and lentiviral depletion using one (notably male) cancer cell line. In line with findings from other tumors (see below), the authors could show that BAG3 expression correlates with cancer cell growth and migration in vitro. Additionally, the authors were able to show that high BAG3 expression protects the cells from cell death induction by conventional chemotherapeutics (5-Fluorouracil), whereas depletion of BAG3 sensitized the cells to treatment und increased apoptosis after treatment. Using a gene expression profiling this study finally showed that BAG3 regulates Interferon, Jak/STAT, AMPK, and PI3K/AKT signaling pathways [43].

### 2.4. Thyroid Cancer

BAG3 was also described to be overexpressed in thyroid cancer samples. Using cell lines, Chiappetta et al. sensitized cancer cells to TRAIL-induced apoptosis-induction via siRNA-depletion of BAG3 [44]. However, another report provides data that depletion of BAG3 in fact induces EMT and that this is accompanied with an increase in ZEB1, a key regulator of EMT, which is mediated by beta-Catenin [45]. A follow-up study from the same group showed that BAG3 expression decreases during starvation of cell lines and that ectopic expression of BAG3 reduced starvation-induced autophagy, although BAG3-overexpression induced autophagy to some extent under normal culture conditions. Additionally, forced expression of BAG3 also resulted in increased apoptosis in BAG3-expressing cells. Conversely, autophagy-inhibition of starved BAG3-expressing cells further induced cell death in these cell lines [46]. Whether these conflicting findings can be resolved in the future is currently unclear. However, future studies might argue for a cell-type dependent function of BAG3 that has hitherto not been described.

### 2.5. Medulloblastoma

In medulloblastoma tissue, BAG3 expression also correlates with tumor grade and shorter patient survival. Interestingly, the study by Yang et al. also determined that BAG3 expression correlates with tumor recurrence. Using Daoy human medulloblastoma cells, they performed a lentiviral depletion of BAG3 and observed reduced migration and invasion, as well as a cell cycle arrest at the S phase [47].

### 2.6. Glioblastoma

Malignant gliomas exhibit a high intrinsic resistance against apoptotic cell death and there is evidence that BAG3 significantly contributes to this phenomenon. We have previously investigated the role of the HSF1/HSP70/BAG3 pathway in resistance of glioma cells to apoptosis-induction with the BH3 mimetics AT-101 and ABT-737 and could demonstrate that pharmacological inhibition of BAG3 (applying the HSF1 inhibitor KRIBB1 and the HSP70/BAG3 interaction inhibitor YM-1) and genetic silencing of BAG3 expression efficiently increased BH3 mimetic-induced cell death and reactivated apoptosis in glioma cells. Depletion of BAG3 also led to a robust loss of cell-matrix adhesion, an almost complete block in the phosphorylation of focal adhesion kinase (FAK) and increased sensitivity to matrix detachment-induced cell death (anoikis). In this context, it is worth mentioning that anoikis can be inhibited by high expression levels of Bcl-2, suggesting that enhanced expression of Bcl-2 may be directly involved in promoting BAG3-dependent anoikis resistance. Furthermore, loss of BAG3 profoundly reduced in vivo tumor growth in an orthotopic mouse glioma model [48]. Similarly, Festa et al. had previously shown that transient depletion of BAG3 via siRNA increases the amount of apoptosis in vitro and strongly reduces tumor growth in vivo using an orthotopic, syngenic rat model (C6 cell line) [49].

Another study addressed the particular role of BAG3 in glioblastoma stem cells (GSCs). It was shown that enriched GSC-like cell cultures exhibit significantly elevated BAG3 mRNA and protein levels, while BAG3 depletion decreased the sphere-forming capacity and decreased expression of the stemness marker SOX-2 and the master regulator of stemness STAT3. Silencing of BAG3 also increased ubiquitination of STAT3, suggesting that STAT3 is another client protein stabilized by BAG3, possibly involved in conferring stem-cell-like properties to GSCs [50].

### 2.7. Ovarian Cancer

Using human ovarian cancer cell lines, Qiu et al. demonstrated that cisplatin-treatment increases autophagy and simultaneous BAG3 expression. Autophagy inhibition sensitized these cells to drug-induced cell death. In line with these findings, genetic depletion of BAG3 lead to an increase of apoptosis and blocked cytoprotective autophagy [51]. A study by Yan et al. revealed that ectopically expressed BAG3 can stabilize the mRNA of the cell cycle regulator Skp2, thus driving cell cycle progression. Furthermore, this study reported that BAG3 antagonizes the function of miR-21-5p, which suppresses Skp2 expression [52].

### 2.8. Liver Cancer

BAG3 is also highly expressed in human hepatocellular carcinoma (HCC), promoting aggressive tumor growth behavior such as invasive growth and neoangiogenesis in these tumors. The study by Xiao et al. showed that BAG3 regulates epithelial-mesenchymal transition (EMT) and angiogenesis in HCC. Silencing of BAG3 evoked a reduction in the migratory and invasive capacity of HCC cells and effectively inhibited tumor growth/metastasis through reduction in CD34 and VEGF expression, events that were associated with a reversion of EMT [53].

In line with these findings, another study showed that BAG3 expression is increased in liver tumors compared to the adjacent tissue, and that BAG3 expression was especially high in tumor with metastases. The same study analyzed the involvement of BAG3 in migration of HCC cells. The authors therefore depleted BAG3 via siRNA and observed reduced migration and invasion and a higher rate of apoptosis, along with a reversal of EMT markers [53].

Unexpectedly, tumor-suppressive functions were also attributed to BAG3 in HCC in a study by Kong et al. [54]. The work showed that ectopically expressed BAG3 could reduce the growth of HCC cell lines in vitro. Mechanistically, it was found that BAG3 suppressed de novo DNA synthesis by inhibiting the pentose phosphate pathway via direct interaction with the rate-limiting enzyme glucose 6 phosphate dehydrogenase (G6PD) [55]. This growth-deficit could further be rescued either by ectopic expression of G6PD or by nucleosides into the culture medium. It should be emphasized that the hypothesis derived from this work relies on ectopic overexpression models, and it remains to be determined whether endogenous BAG3 can really exert tumor-suppressive functions in certain cases, while in most observed cases, BAG3 rather displays a tumor-promoting function.

### 2.9. Lung Cancer

It was found that BAG3 is also frequently overexpressed in different types of lung cancer, including small cell lung carcinomas (SCLC). Chiapetta et al. investigated the expression of BAG3 in >60 specimens from different lung tumors and analyzed the role of this protein in SCLC cell death resistance. Indeed, depletion of BAG3 in two human SCLC cell lines evoked increased cell death and sensitized cells to treatment with the chemotherapeutic agent cisplatin. BAG3 silencing also reduced tumor growth in an in vivo xenograft model of SCLC. The authors conclude that a subset of SCLCs overexpress BAG3, thereby increasing resistance to chemotherapy [56].

### 2.10. Cervical Cancer

In cervical cancer, BAG3 expression was shown to correlate with tumor grade [57,58]. Additionally, siRNA-mediated depletion of BAG3 inhibits epithelial-mesenchymal transition (EMT) in cervical cancer cell lines in vitro and reduced tumor sizes in xenograft mouse model in vivo. This was accompanied by reduction of the EMT-regulator Slug and N-Cadherin and MMP2 as marker protein for the mesenchymal state and an increase in E-Cadherin, indicating an epithelial state [58].

### 2.11. Rhabdomyosarcoma

Rapino et al. tested whether combined inhibition of the ubiquitin-proteasomal-system (UPS) using Bortezomib and the aggresome-autophagy pathway using the HDAC6-inhibitor ST80 could be employed to target these cancer cells by abrogating both protein quality control mechanisms. Interestingly, they observed that in the surviving fraction of cells, BAG3 was highly expressed. Lentiviral depletion of BAG3 could revert these pro-survival effects and BAG3 was further shown to be necessary for the removal of protein aggregates. Similar findings were achieved after ATG7-knockdown and late-stage autophagy-inhibition using BafA1, indicating that the increase in BAG3 expression drives a compensatory mechanism that involves autophagy [25].

### 2.12. Melanoma

Franco et al. reported that BAG3 is expressed in a proportion of primary melanomas and in the majority of melanoma metastases, suggesting a potential role for this protein in tumor development [59]. In an in vivo model of human melanoma, *BAG3* silencing resulted in a significant reduction in tumor growth and prolonged animal survival [11]. The same study also demonstrated that BAG3 serves to sustain protein levels of IKK-γ in melanoma cells, thereby allowing the continuous activation of the anti-apoptotic NF-ĸB pathway.

### 2.13. Pancreatic Cancer

Rosati et al. analyzed tumor samples via IHC and qRT-PCR and demonstrated that high intracellular BAG3 expression correlates with shorter survival [60]. Using human cell lines, they further showed that treatment with gemcitabine increases BAG3 mRNA levels and that siRNA-mediated depletion of BAG3 induced a G0/G1 cell cycle arrest. Additionally, Falco et al. proposed that BAG3 is a novel serum biomarker for pancreatic adenocarcinomas [61].

The study by An et al. investigated the potential role of BAG3 in metabolic reprogramming of PDAC. BAG3 was shown to increase expression of Hexokinase 2, a key enzyme involved in glycolysis (an event driven by interaction of BAG3 with HK2 mRNA). BAG3 expression levels were associated with recruitment of the RNA-binding proteins Roquin and IMP3 to the HK2 mRNA. The authors proposed that in PDAC, BAG3 promotes reprogramming of glucose metabolism via interaction with HK2 mRNA [62].

Another study showed that BAG3 indirectly regulates IL-8 expression and thereby regulates the migration and invasion of PDAC cells. The proposed mechanism is that BAG3-depletion prevents the cytosolic translocation of the nuclear protein Human antigen R (HuR), thereby preventing it from stabilizing IL-8 mRNA, while simultaneously promoting the binding of miR-4132 to Ago2, which in turn degrades the IL-8 transcript [63]. These two studies nicely demonstrate that the pleiotropic functions of BAG3 are not only limited to protein-protein-interactions, but that BAG3 also has profound impacts on mRNA stability, which should be taken into consideration when analyzing global approaches like transcriptomics or proteomics.

Interestingly, Rosati et al. found that PDAC cells secrete BAG3 into the extracellular space, where BAG3 binds to macrophages, inducing their activation and the secretion of factors supporting PDAC growth. The authors went on to identify IFITM-2 (Interferon-induced transmembrane protein 2) as a BAG3 receptor signaling through the PI3K and p38 MAPK pathways. They also demonstrated that a neutralizing anti-BAG3 antibody reduced tumor growth and prevented metastasis formation in three different mouse models. This study identified a novel paracrine loop involved in PDAC growth and metastatic spreading, and suggested that-BAG3 antibodies carry therapeutic potential for the treatment of PDAC [8,64].

Furthermore, Yuan et al. recently showed that the tumor stroma, pancreatic stellate cells, occasionally expresses high levels of BAG3. Using ectopic overexpression of BAG3 in vitro in pancreatic stellate cells, they could further exemplify that conditioned medium derived from these cells is capable to induce migration and invasion in PDAC cell lines and that this was mediated by expression and secretion of cytokines like IL-8, TGF-β2 and IGFBP2 [65].

## 3. Outlook

Although the HSP70/BAG3 interaction inhibitors YM-1 and the structurally related JG-98 are already available [12,66], they have to be used in micromolar concentrations to achieve effective BAG3 inhibition. Therefore, off-target based mechanisms of these compounds cannot be excluded to be involved in their antiproliferative and pro-apoptotic effects, and their lack of sufficient specificity limits their translational potential. To pharmacologically target BAG3 in the clinical setting, it will be of crucial importance to develop better, more specific HSP70 binders preventing the HSP70/BAG3 interaction, or to develop high-affinity direct BAG3 inhibitors in the future. Of note, the first-of-its-kind direct chemical BAG3 inhibitor was recently described by Terracciano et al. The authors employed a virtual screening approach, followed by experimental validation and chemical modification to report a 2,4-thiazolidinedione derivative, which could be used as a scaffold for the further development of a potent chemical BAG3 inhibitor [67].

A second approach to target BAG3 in the clinical setting is the use of neutralizing antibodies. Based on the findings of Rosati et al. [64], which demonstrate that tumor growth and metastasis is partially blocked with an anti-BAG3 murine antibody, secreted BAG3 may represent a potential target for the treatment of pancreatic cancer. In follow-up work, Basile et al. were able to generate a humanized version of this neutralizing anti-BAG3 antibody (BAG3-H2L4) that is currently developed for clinical application [68]. In their published work, the authors were able to show that BAG3-H2L4 prevents attachment of BAG3 to macrophages and their release of the cytokine IL-6. BAG3-H2L4 also significantly inhibited the growth of Mia PaCa-2 xenografts and was found to be enriched in tumor tissues. Based on these results, secreted BAG3 (and its receptor IFITM-2) represent potential clinical candidate molecules for the treatment of pancreatic cancer (development of an anti-BAG3 humanized antibody for treatment of pancreatic cancer). Clearly, pancreatic ductal adenocarcinoma is the likeliest candidate for the first clinical investigations of BAG3-directed therapies.

Given the relevance of BAG3 for many other types of cancer, cancer cell proliferation, and therapy and stress resistance, other tumor entities may follow this line of research in the near future and cancer research may add BAG3 to the list of potential therapeutic and diagnostic markers.

## Figures and Tables

**Figure 1 cells-09-00574-f001:**
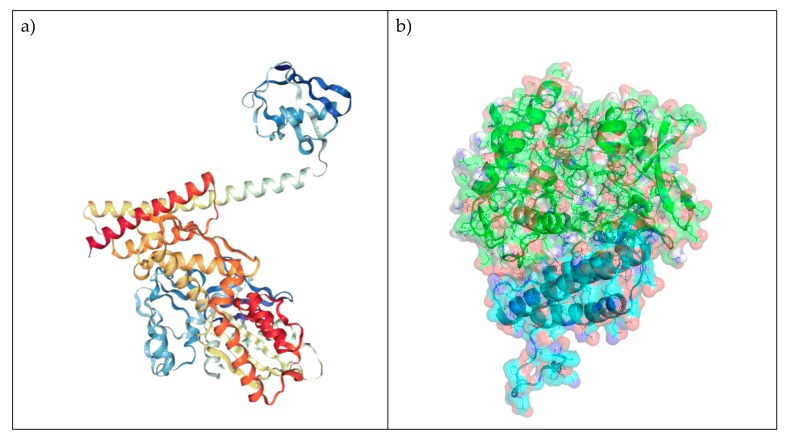
Modelled interaction of BAG3 with HSP70. (**a**) Crystal structure of the homologue BAG1 binding to HSP70 (PDB 4HWI). (**b**) Homology model of BAG3 (cyan) binding to HSP70 (green). (**c**) Superimposition of BAG1 (magenta) and BAG3 (cyan) in complex with HSP70 (green). (**d**) Two-dimensional and three-dimensional depiction of plausible key interactions observed by visual inspection of the modelled interaction, BAG3 residues are coloured in light blue and HSP70 residues in green.

**Figure 2 cells-09-00574-f002:**
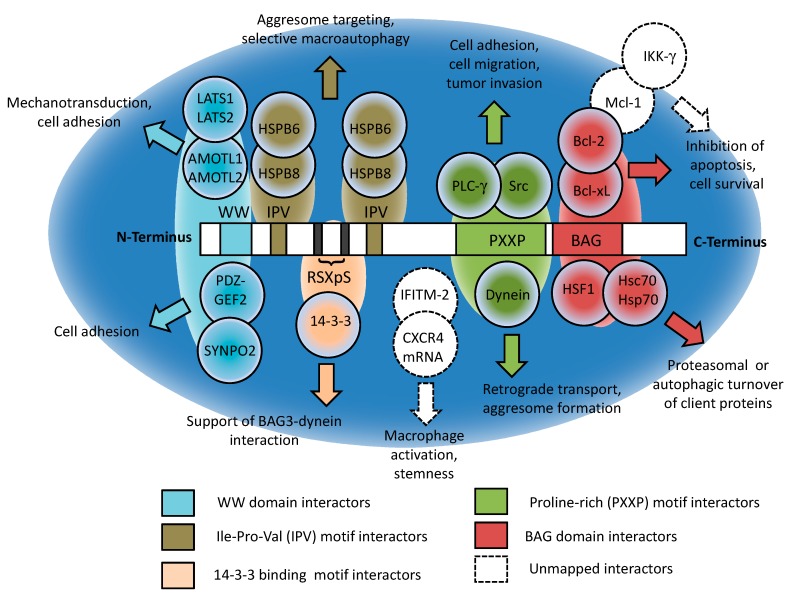
Domain structure of BAG3 and major interactors: BAG3 as a hub molecule regulating multiple pathways related to stress and therapy resistance. Key functional/interaction domains of BAG3, major BAG3 interactors, and associated pathways are depicted. The WW domain of BAG3 binds to proline-rich repeats of interactors, including LATS1/2, AMOTL1/2, the guanine nucleotide exchange factor 2 (PDZGEF2), and synaptopodin-2 (SYNPO2). The two conserved IPV motifs mediate complex formation of BAG3 to the small heat shock proteins HSPB8 and HSPB6. The PxxP motif interacts with the SH3 (Src homology 3) domains of phospholipase C gamma (PLC-γ), Src and the motor protein dynein. BAG3 also possesses two phosphoserine-containing 14-3-3 binding motifs that interact with the 14-3-3γ protein. The C-terminal BAG domain of BAG3 binds to several interactors, including the anti-apoptotic BCL-2 protein, the ATPase domain of the HSC/HSP70 chaperone, and the heat shock factor and BAG3 upstream transcriptional regulator HSF1. For further information, please refer to the main text and to [3]. WW: WW domain; IPV: Ile-Pro-Val motif; RSXpS: 14-3-3 binding motif; PXXP: proline-rich motif; BAG: BAG domain.

**Table 1 cells-09-00574-t001:** Aberrant expression of BAG3 in different cancer entities.

Tumor Entity	Findings	Reference
Cervix cancer	BAG3 expression correlates with grade	[57]
Chondrosarcoma	Higher BAG3 expression in tumor tissue compared to normal cartilage tissue and benign tumor chondroma	[69]
Chronic Lymphocytic Leukemia (CLL)	BAG3 expression increased in tumor and high BAG3 expression correlates with shorter survival	[39]
Colon Cancer	Higher expression compared to adjacent non-tumor tissue; females have higher BAG3 expression; large tumors (>5 cm) have higher BAG3 expression	[43]
Endometrioid endometrial adenocarcinomas	Higher expression of BAG3 compared to adjacent healthy tissue	[70]
Glioma	BAG3 expression correlates with tumor grade	[49]
Hepatocelllular Carcinoma (HCC)	Higher expression of BAG3 in tumor compared to adjacent tissue; high BAG in metastases and higher BAG3 expression in Grade III–IV tumors	[53]
Lung Cancer	Overexpression of BAG3 in different types of lung cancer (squamous cell carcinomas, adenocarcinomas, large cell carcinomas and small cell lung cancer)	[56]
Medulloblastoma	BAG3 expression correlates with tumor grade; BAG3 expression correlates with recurrence; BAG3 expression correlates with shorter survival	[47]
Melanoma	No difference in BAG3 intensity (IHC) in primary tumors; higher expression in lymph node metastases; highest expression in organ metastases	[59]
Pancreatic Ductal Adenocarcinoma (PDAC)	High BAG3 expression correlates with shorter survival	[60]
Testicular Cancer	No difference between normal and tumor tissue, but higher BAG3 expression in seminoma compared to non-seminoma tumors; higher BAG3 and p62 expression in perivascular areas and in proximity to necrotic foci	[71]
Thyroid Cancer	Little to no expression in normal tissue; high expression in most tumor samples	[44]

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
