# Peer review of "At the Crossroads of Apoptosis and Autophagy: Multiple Roles of the Co-Chaperone BAG3 in Stress and Therapy Resistance of Cancer"

_cells, 2020, doi:10.3390/cells9030574_

Round 1

Reviewer 1 Report

This article reviews the multiple functions of BAG3, an Hsp70-bound co-chaperone that works as a hub protein connecting several biological events ranging from cellular stress response, proteostasis and cell death regulation to cell development, cytoskeletal dynamics and cancer.

The article addresses several aspects related to the involvement of BAG3 in cell death, cell stress, and cancers. It is well written and each topic acceptably developed. References are appropriated.

In view of the relevance of the BAG3-HSP70 interaction, I would suggest including a specific section related to this specific association, from the structural and biological perspectives. An integrative figure of the complex showing the interacting domains of these proteins (either a simple scheme or even better, a molecular modelling figure) would be useful as a graphical reference for a broad number of readers interested in the field.

In Fig.1, it is shown an important association of BAG3 with the motor protein dynein via the PXXP domain. In the texts, there is no comment related to this relevant interaction nor supporting references are cited. In view of the number of client proteins able to associate to BAG3, the potential role of dynein as the molecular motor responsible for the retrotransport of those heterocomplexes should be addressed in a specific section.

Author Response

This article reviews the multiple functions of BAG3, an Hsp70-bound co-chaperone that works as a hub protein connecting several biological events ranging from cellular stress response, proteostasis and cell death regulation to cell development, cytoskeletal dynamics and cancer. The article addresses several aspects related to the involvement of BAG3 in cell death, cell stress, and cancers. It is well written and each topic acceptably developed. References are appropriated.

We thank the reviewer for this appreciation.

In view of the relevance of the BAG3-HSP70 interaction, I would suggest including a specific section related to this specific association, from the structural and biological perspectives. An integrative figure of the complex showing the interacting domains of these proteins (either a simple scheme or even better, a molecular modelling figure) would be useful as a graphical reference for a broad number of readers interested in the field.

We have now prepared a new text (page 3, 1st paragraph, marked in yellow) and new Figure (new Fig. 1) as requested. Since a crystal structure of the BAG3-Hsp70 complex has to date not been published, we used the crystal structure of the homologous BAG1-Hsp70 complex to create a homology model. Further details can be found in the new Results section (page 3, 1st paragraph, marked in yellow) and Figure legend.

In Fig.1, it is shown an important association of BAG3 with the motor protein dynein via the PXXP domain. In the texts, there is no comment related to this relevant interaction nor supporting references are cited. In view of the number of client proteins able to associate to BAG3, the potential role of dynein as the molecular motor responsible for the retrotransport of those heterocomplexes should be addressed in a specific section.

This is a very good point. We have included this information (page 7, end of second paragraph, marked in yellow) and the supporting references now in the revised manuscript.

Reviewer 2 Report

My suggestions for the authors are as follows:

1) include lung cancer obtained evidences in table 1.

2) check on lanes 336-338:  Falco et al. reference is not in the list. Furthermore, the cited report refers to investigations on extracellular  BAG3 (detection in patients' sera)  and not on intracellular BAG3 as mentioned by authors.

3) a paragraph on evidences obtained in melanoma (BAG3 expression in human tumours and impact of BAG3 expression/activity modulation in vitro and in vivo) could be add in this review.

Author Response

My suggestions for the authors are as follows:

1) include lung cancer obtained evidences in table 1.

The requested information on lung cancer has been incorporated into the revised table.

2) check on lanes 336-338:  Falco et al. reference is not in the list. Furthermore, the cited report refers to investigations on extracellular  BAG3 (detection in patients' sera)  and not on intracellular BAG3 as mentioned by authors.

Thank you for bringing this mistake to our attention. The text and the citations have been rearranged accordingly (page 17, 4th paragraph ff, marked in yellow).

3) a paragraph on evidences obtained in melanoma (BAG3 expression in human tumours and impact of BAG3 expression/activity modulation in vitro and in vivo) could be add in this review.

This is an excellent suggestion. A new paragraph on BAG3 in melanoma has been added now (Page 17, third paragraph of the revised manuscript, marked in yellow).

Reviewer 3 Report

The manuscript by Dr. Kogel et al., provides an exhaustive list of the multiple roles of BAG3 in different type of cancer. It is well written and the literature has been well reviewed however the “Crossroads of apoptosis and autophagy” as mentioned in the title will benefit from more highlights in the text. According to the title the reader would anticipate to read more about apoptosis and autophagy and how BAG3 balances those two key pathways. The apoptosis is described in the introduction but not linked or discussed in the context of autophagy, while autophagy is briefly mentioned at the beginning of the introduction. The manuscript will greatly benefit from a full section on the role of BAG3 in apoptosis/autophagy illustrated by figures. 

Minor comments:

Line 15 need space after full stop

Line 33 figure 1 should be mentioned

Line 220 PBMCs

Line 237 Additionally the authors “were”

Line 274 space

Lines 287-290 are in bold?

Line 357 font size

Author Response

The manuscript by Dr. Kogel et al., provides an exhaustive list of the multiple roles of BAG3 in different type of cancer. It is well written and the literature has been well reviewed however the “Crossroads of apoptosis and autophagy” as mentioned in the title will benefit from more highlights in the text. According to the title the reader would anticipate to read more about apoptosis and autophagy and how BAG3 balances those two key pathways. The apoptosis is described in the introduction but not linked or discussed in the context of autophagy, while autophagy is briefly mentioned at the beginning of the introduction. The manuscript will greatly benefit from a full section on the role of BAG3 in apoptosis/autophagy illustrated by figures. 

The possible crosstalk between apoptosis and autophagy certainly is a very important point, but has so far not been that well studied in the direct context of BAG3 and malignant disease. We are confident that the review covers all currently available and relevant studies on BAG3-dependent modulation of apoptosis and autophagy in cancer and we have now integrated a new discussion on the possible function of BAG3 in regulating the apoptosis/autophagy crosstalk based on the currently available knowledge (page 7, third paragraph ff, marked in yellow). In general, an extensive cellular crosstalk between the apoptosis and autophagy pathways involving dual regulators of both pathways (e.g. ATG5, ATG12, p53) is well established, so modulation of apoptosis by BAG3 should also affect autophagy to some extent and vice versa. Based on this concept and the fact that BAG3 is a mutimodal hub molecule it could be argued that its roles in modulation of apoptosis and autophagy are separate, distinct functions of the protein promoted by its specific interactors. Potential additional mechanisms linking apoptosis and autophagy signaling (directly) at the level of BAG3 remaining to be defined. We also elaborate on a new hypothesis stating that the anti-apoptotic vs pro-autophagic activity of BAG3 may be driven by the respective cellular demand. Under conditions of severe misfolded protein accumulation (as induced by many types of cancer treatment), there may be a shift from BAG3-dependent apoptosis inhibition towards more autophagy activation, possibly involving most/all of the cellular pool of BAG3 in this setting, with a possible impact on the response to therapy.

The established BAG3-interactors relevant for apoptosis and autophagy are depicted in Fig. 2 (old Fig. 1) and we therefore decided not to prepare a separate, additional Figure on this topic.

Minor comments:

Line 15 need space after full stop

Line 33 figure 1 should be mentioned

Line 220 PBMCs

Line 237 Additionally the authors “were”

Line 274 space

Lines 287-290 are in bold?

Line 357 font size

All these mistakes have been corrected now.

Round 2

Reviewer 1 Report

The authors have properly addressed all reviewers' concerns. I feel that the article could be published in its present form.

Reviewer 3 Report

Thank you to the authors for addressing my different comments.